# *Paraclostridium tenue* Causing an Anaerobic Brain Abscess Identified by Whole-Metagenome Sequencing: A Case Report

**DOI:** 10.3390/microorganisms12081692

**Published:** 2024-08-16

**Authors:** Tetsuya Chiba, Yorito Hattori, Daisuke Motooka, Tomotaka Tanaka, Masafumi Ihara

**Affiliations:** 1Department of Neurology, National Cerebral and Cardiovascular Center, Suita, Osaka 564-8565, Japantanakat@ncvc.go.jp (T.T.);; 2Department of Preemptive Medicine for Dementia, National Cerebral and Cardiovascular Center, Suita, Osaka 564-8565, Japan; 3Department of Infection Metagenomics, Research Institute for Microbial Diseases, Osaka University, Suita, Osaka 565-0871, Japan

**Keywords:** brain abscess, *Paraclostridium tenue*, whole-metagenome sequencing

## Abstract

When treating anaerobic brain abscesses, healthcare professionals often face the difficulty of identifying the causal pathogens, necessitating empiric therapies with uncertain efficacy. We present the case of a 57-year-old woman who was admitted to our hospital with a fever and headache. Brain magnetic resonance imaging revealed a hemorrhagic lesion with wall enhancement at the left hemisphere on contrast-enhanced T1-weighted imaging. Cerebrospinal fluid examination showed pleocytosis (23 cells/μL), an elevated protein level (125 mg/dL), and decreased glucose level (51 mg/dL; blood glucose was 128 mg/dL). Intracerebral hemorrhage accompanied by a brain abscess was clinically suspected. The patient received empirical treatment with intravenous meropenem and vancomycin for 2 weeks. However, conventional bacterial culture tests failed to identify the pathogen. We then performed shotgun sequencing and ribosomal multilocus sequence typing, which identified *Paraclostridium tenue*. Based on this finding, we de-escalated to benzylpenicillin potassium for 4 weeks, leading to a 2.5-year remission of the anaerobic brain abscess. Therefore, *Paraclostridium* can be a causative pathogen for brain abscesses. Furthermore, whole-metagenome sequencing is a promising method for detecting rare pathogens that are not identifiable by conventional bacterial culture tests. This approach enables more targeted treatment and contributes to achieving long-term remission in clinical settings.

## 1. Introduction

A brain abscess is a serious medical condition requiring immediate attention due to high mortality and unfavorable neurological outcomes. It can be treated using antimicrobial therapy and surgical drainage. Over the past five decades, the prognosis for brain abscesses has improved due to technical advancements, such as the modality of cranial imaging, neurosurgical techniques, and improved antimicrobial regimens. A systematic review showed that the fatality rate has decreased from 40% to 10%, and the rate of full recovery has increased from 33% to 70% [1]. However, a nationwide study in Denmark from 2010 to 2016 still showed a mortality rate of 20% [2], and insufficient treatment may induce epilepsy, with 32% of patients experiencing a new development of epilepsy as the most common neurologic sequela [3,4]. Therefore, it is crucial to accurately identify the pathogen causing brain abscesses and administer appropriate antibiotics targeting etiological bacteria.

Furthermore, diagnosing anaerobic abscesses is particularly challenging due to the complexities involved in isolating anaerobic bacteria. Invasive brain biopsies may occasionally be performed to confirm the diagnosis and identify the causal pathogen. In our case, pathogenetic microorganisms could not be identified by conventional bacterial culture tests initially. However, whole-metagenome sequencing finally identified *Paraclostridium tenue* in cerebrospinal fluid (CSF) collected by lumbar puncture, and the long-term remission of the anaerobic brain abscess was achieved through the appropriate antibiotic therapy. This approach helps to ameliorate infection, prevent recurrence, and ultimately achieve sustained remission. Therefore, whole-metagenome sequencing is valuable for identifying pathogens and achieving long-term remission and should be more commonly employed in daily clinical settings.

## 2. Materials and Methods

We used a test-tube growth medium for anaerobic bacteria and a test-tube growth medium supplemented with hemin and vitamin K_1_. The enriched bacteria were then transferred to an agar medium suitable for anaerobic culture, which was also supplemented with hemin and vitamin K_1_, and incubated at 35 °C to obtain pure cultures.

To identify the causative microorganism, we performed the 16S rRNA gene and whole-metagenome sequencing of the pure cultured bacteria from the cultured strains of CSF. First, we used MEGA (https://www.megasoftware.net/ (accessed on 15 August 2024)) for the phylogenetic analysis of the full-length 16S rRNA gene sequence. Subsequently, whole-metagenome analysis was performed using DNA extracted from cultured bacterial isolates. The genome sequence was assembled into 28 contigs by shotgun sequencing using MiSeq^®^ (Illumina, San Diego, CA, USA) and de novo assembly by Unicycler (ver. 0.4.4). Ribosomal multilocus sequence typing (rMLST, https://pubmlst.org/species-id (accessed on 15 August 2024)) was used as well [5]. Finally, the genomic information was compared to the other two strains with previously reported whole metagenomes. Additionally, we used whole-metagenome information for average nucleotide identity (ANI) analysis (https://github.com/ParBLiSS/FastANI (accessed on 15 August 2024)) [6].

## 3. Case Description

A 57-year-old woman presented at our hospital with a complaint of fever and headache lasting for several days. She had a history of embolic stroke of an undetermined source and took warfarin (4.25 mg/day). Her body temperature was 37.7 °C. The neurological examinations revealed alert consciousness and mild hearing loss in the left ear, with no other neurological abnormalities such as meningeal irritation signs and motor palsy. Blood tests showed an elevated white blood cell count of 5, 180/μL, CRP 0.04 mg/dL and PT-INR 2.89. The head non-contrast computed tomography (CT) showed a circular hemorrhagic lesion with an air-fluid level inside at the left hemisphere, attaching to the left ventricle (Figure 1A). Brain magnetic resonance imaging (MRI) showed a hypointense round lesion on diffusion-weighted imaging (Figure 1B), a cystic lesion with heterogeneous hypo- and hyper-intensities on fluid-attenuated inversion recovery imaging (Figure 1C) and T2*-weighted imaging (Figure 1D), a lesion wall with hyperintensity on T1-weighted imaging (Figure 1E), and the enhancement of the wall on contrast enhancement T1-weighted imaging (Figure 1F). MR angiography (Figure 1G), as well as conventional cerebral angiography, showed no abnormal vessels or aneurysms. A CSF examination showed that the cell count was mildly elevated (23/μL; polynuclear cells 5/μL: mononuclear cells 18/μL), and the protein level was also elevated (125 mg/dL), but the glucose level was decreased (51 mg/dL; blood glucose of 128 mg/dL examined at the same time). According to the findings indicated above, a brain abscess accompanied by intracerebral hemorrhage was clinically suspected.

CSF was further anaerobically cultured prior to treatment, which indicated the existence of *Clostridium* spp. with suspicion; thereby, the detailed pathogenetic microorganism was not identified by a conventional bacterial culture test. No sinusitis or inflammatory diseases of the inner or middle ear were present in the head CT. A thoracoabdominal contrast-enhanced CT showed no obvious inflammatory lesions. Transthoracic echocardiography showed no vegetation on valves. A dental examination revealed no caries. The patient was diagnosed with a brain abscess accompanied by an intracerebral hemorrhage and started to be empirically treated with intravenous meropenem 6 g/day and vancomycin 3 g/day for 2 weeks.

To identify the causative microorganism, we performed 16S rRNA gene and whole-metagenome sequencing of the pure cultured bacteria from the cultured strains of CSF. First, phylogenetic analysis of the full-length 16S rRNA gene sequence analysis revealed that the bacteria were close to strains of *P. tenue*, *Paeniclostridium ghonii*, and *Paeniclostridium sordellii* (Figure 2). Subsequently, we performed whole-metagenome analysis using DNA extracted from cultured bacterial isolates. The genome sequence was assembled into 28 contigs by shotgun sequencing using MiSeq^®^ and de novo assembly by unicycler. Ribosomal multilocus sequence typing showed that *P. tenue* was identified. Finally, we scrutinized its genomic information to determine whether whole genomes of the identified *P. tenue* were equivalent to other strains of *P. tenue* for which whole metagenomes have been previously reported [7]. A similar genome size, number of genes, and GC% among the two strains of *P. tenue* were observed (Table 1). Furthermore, we used whole-metagenome information to perform an ANI analysis of the strains of *P. tenue*, *P. sordellii* and *P. ghonii*. As a result, we confirmed that the bacterium responsible for the brain abscess was *P. tenue*, not *P. sordellii*, and *P. ghonii* (Table 2). Therefore, we determined that biopsy and surgery were not feasible or necessary for the patient as their symptoms were limited to fever, headache, and mild hearing loss without consciousness disturbances or motor palsy. Additionally, *P. tenue* was identified as the causal pathogen through shotgun sequencing. Finally, we de-escalated from broad-spectrum antibiotics to narrow-spectrum antibiotics, benzylpenicillin potassium 24,000,000 U/day for 4 weeks. The brain abscess was found to be shrunken in two months and has not been exacerbated or undergone relapse over 2.5 years (Figure 3).

## 4. Discussion and Conclusions

*P. tenue* was identified as the causative microorganism, in this case, as a member of the genus *Paraclostridium*, which is a saccharolytic anaerobic Gram-positive rod. This bacterium is a member of the human gut microflora and can be isolated from human fecal samples [8]. There are few reports of *P. tenue* infections, including bacteremia, due to difficulties identifying it by a conventional bacterial culture test. Therefore, *P. tenue* is generally required to be identified by metagenomic analysis [9].

Patients who have intestinal comorbidities such as colon cancer are susceptible to bacteremia caused by Gram-positive rods, including *Paraclostridium* [10]. In the development of colorectal cancer, inflammation and oxidative stress in the colon epithelial cells are thought to be involved in colorectal carcinogenesis, leading to the breakdown of the integrity of the colonic barrier [11]. Furthermore, abundant *Paraclostridium* may inhibit expression of proteins constituting tight junction and adherent junction, and anti-inflammatory cytokines (interleukin-10), by contrast activate proinflammatory cytokines (interleukin-6, tumor necrosis factor-α and interferon-γ) in the colon may be increased [12]. Such mechanisms may have facilitated intestinal barrier disruption, allowing enteric commensal bacteria to encroach in the intestinal wall and microcirculation [11]. In this case, the intestinal barrier could have been disrupted, and *P. tenue* could have invaded the luminal wall and caused hematogenous bacterial translocation to the brain.

Generally, patients with brain abscesses may require stereotactic-guided needle aspiration or surgical drainage in addition to antibiotics for both diagnostic and therapeutic purposes. However, for this patient, these procedures were deemed too invasive given the mild nature of her symptoms, which included only fever, headache, and mild hearing loss without consciousness disturbances or motor palsy. Moreover, the identification of *P. tenue* as the causal pathogen by shotgun sequencing using CSF made these invasive procedures unnecessary, leading to a successful de-escalation from empirical therapy.

Similar to this case, only 24% of conventional CSF cultures identify the causative microorganism of brain abscesses [1]. Brain abscesses caused by anaerobic bacteria are especially difficult to diagnose because of the difficulty of isolation techniques. Thus, next-generation sequencing, such as metagenomic analysis, is useful for the elucidation of new pathogens. To date, whole-metagenomic analysis has identified causal pathogens in 17 cases of brain abscesses, while bacterial culture tests have identified pathogens in only six of these cases (Table 3, [13,14,15,16,17,18,19,20,21,22,23,24,25,26]). In our case, we were able to identify *P. tenue* as the causative microorganism via the 16S rRNA gene and whole-metagenome sequencing. In 16S rRNA analysis, the 16S rRNA gene of a bacterium is amplified by PCR, sequencing analysis is performed, and a homology comparison with known databases is performed to narrow down the genus and species names. Conversely, whole-metagenome analysis is a very useful method to obtain genetic composition because it reads all the fragmented gene sequences without selecting specific regions using PCR. In ANI analysis, the whole-metagenome sequence is logically cut into about 1000 bases, and the similarity to all the sequences of the partner is calculated. Within the same species, the ANI index is greater than 95% [27], as this case showed a similar percentage (Table 2).

An intracerebral hemorrhage accompanied by a brain abscess was clinically suspected in this case. Intracerebral hemorrhages are rarely involved in brain abscesses. The mechanism of a hemorrhage in the brain abscess is still unclear, but neoangiogenesis observed in brain abscesses was demonstrated by the increased expression of vascular endothelial growth factor and microvascular density [28]. The neoangiogenesis in the wall of the brain abscesses is susceptible to the rupture of newly formed fragile blood vessels [29]. In our case, contrast-enhanced T1-weighted imaging revealed an enhanced abscess wall, suggesting the presence of neovascularization in the wall. Furthermore, the patient was taking warfarin, which could contribute to the bleeding tendency.

The patient experienced mild hearing loss. CSF analysis revealed the presence of *P. tenue.* This finding indicates that *P. tenue* in the CSF was the etiological agent responsible for bacterial meningitis in this case. Meningitis caused by *P. tenue* can lead to sensorineural hearing loss, a common complication of bacterial meningitis, affecting up to 54% of survivors [30]. Several experimental studies have shown that lesions can include suppurative labyrinthitis with cochlear damage and infection that spreads from the subarachnoid space [31,32,33,34,35].

In conclusion, *P. tenue* was considered a causative microorganism of the anaerobic brain abscess in this case, as whole-metagenome sequencing demonstrated its presence. Whole-metagenome sequencing is a promising method for detecting rare pathogens that cannot be identified by conventional bacterial culture tests commonly used in clinical settings and research and for achieving the long-term remission of brain abscesses.

## Figures and Tables

**Figure 1 microorganisms-12-01692-f001:**
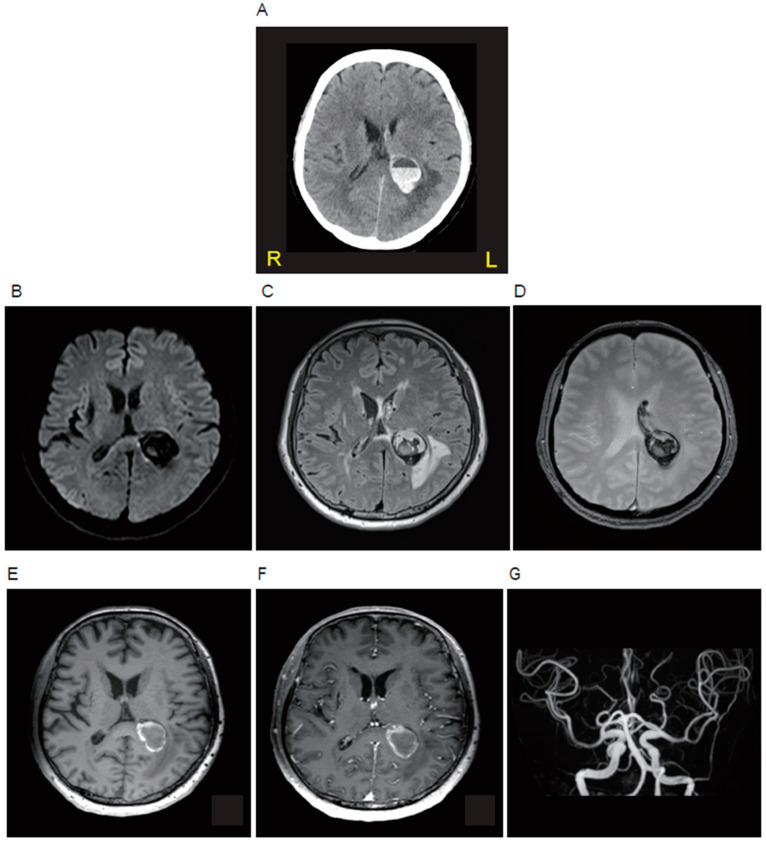
Neuroradiological findings of the patient in the acute phase. The brain lesion is shown by brain computed tomography (**A**), diffusion-weighted magnetic resonance imaging (MRI) (**B**), fluid-attenuated inversion recovery MRI (**C**), T2*-weighted MRI (**D**), T1-weighted MRI (**E**), contrast-enhanced T1-weighted MRI (**F**), and MR angiography (**G**).

**Figure 2 microorganisms-12-01692-f002:**
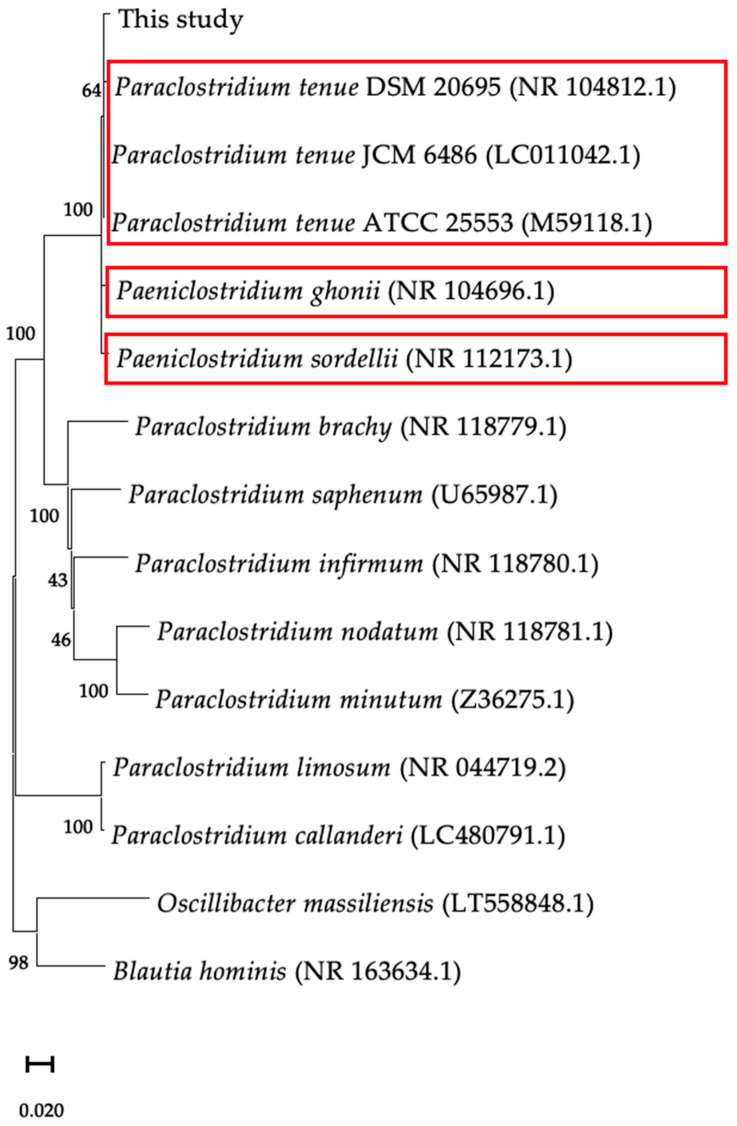
Phylogenetic tree constructed using 16S rRNA sequences. A phylogenetic tree was constructed using 16S rRNA sequences of closely related (red square frame) and distantly related species (*Oscillibacter* and *Blautia*).

**Figure 3 microorganisms-12-01692-f003:**
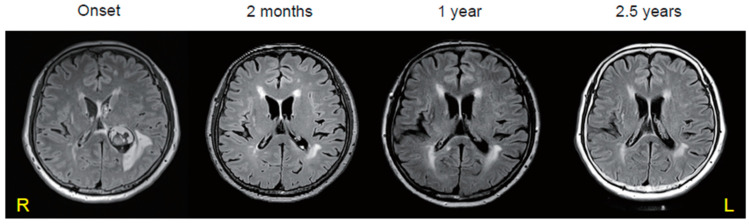
Long-term follow-up of brain magnetic resonance imaging. Fluid-attenuated inversion recovery imaging examined at the disease onset and after 2 months, 1 year, and 2.5 years.

**Table 1 microorganisms-12-01692-t001:** Genome assembly statistics between *Paraclostridium tenue* strains.

	This Study	JCM 6486 [7]
Isolation source	human CSF	Post-abortion abscess
Number of contigs	28	64
N50 contig length [bp]	263,515	163,012
Max contig length [bp]	864,437	342,076
No. of bases, total [bp]	2,995,440	2,943,107
GC content [%]	27	26.9
Number of CDSs	2963	2875

Abbreviations: CDS, coding sequence; CSF, cerebral spinal fluid; and GC, guanine–cytosine.

**Table 2 microorganisms-12-01692-t002:** Comparison of average nucleotide identity (ANI).

	*Paraclostridium tenue* JCM6486 (GCA_039521565.1)	*Paraclostridium sordellii* ATCC9714 (GCA_000953675.1)	*Paraclostridium ghonii* DSM15049 (GCA_030815085.1)
This case	95.1	85.3	83.4
*Paraclostridium tenue* JCM6486 (GCA_039521565.1)	NA	85.1	83.4
*Paraclostridium sordellii* ATCC9714 (GCA_000953675.1)	NA	NA	82.6

Abbreviation: NA, not applicable.

**Table 3 microorganisms-12-01692-t003:** Reported cases of brain abscesses diagnosed by whole-metagenomic analysis.

References	Age (Years)	Specimen	Antibiotics Therapy Prior to Sampling	Culture Results	16S rRNA Analysis Results	Whole-Metagenomic Analysis	Whole-Metagenomic Analysis Results
Our case	57	CSF	No	Negative	*P. tenue*, *P. ghonii*, *P. sordellii*	SMg	*P. tenue*
[13]	75	CSF	No: prior to culture, Yes: prior to metagenomic analysis	Negative	NA	SMg	*Prevotella* *denticola*, *Fusobacterium* *nucleatum*
[14]	50	CSF	No: prior to culture, Yes: prior to metagenomic analysis	Negative	NA	SMg	*Nocardia* *farcinica*
[15]	58	CSF	Yes	Positive	NA	SMg	*N. farcinica*
[16]	65	CSF	Yes	Negative	NA	SMg	*Porphyromonas* *gingivalis*
[17]	49	CSF	Yes	Positive	NA	SMg	*N. brevicatena*
[18]	30	CSF	Yes	Positive	NA	SMg	*Scedosporium* *boydii*
[19]	67	CSF	Yes	Negative	NA	SMg	*Streptococcus* *intermedia*, *N. asiatica*
[20]	66	CSF	Yes	Negative	NA	SMg	*S. suis*
[21]	2 months	Pus from brain abscess	Yes	*Bacteroides* *fragilis*	NA	SMg	*Bacteroides* *fragilis*
[21]	5	Pus from brain abscess	Yes	*S. intermedius*	NA	SMg	*S. intermedius,* *S. constellatus*
[21]	10	Pus from brain abscess	Yes	Negative	NA	SMg	*S. intermedius*
[21]	13	Pus from brain abscess	Yes	Negative	NA	SMg	*P. oris*, *F. nucleatum*, *S. intermedius*
[22]	61	Pus from brain abscess	Yes	Negative	NA	SMg	*N. asiatica*
[23]	27	Pus from brain abscess	Yes	Negative	NA	SMg	*P. loescheii*
[24]	70	Pus from brain abscess	Yes	Negative	Negative	SMg	*S. anginosus*, *F. nucleatum*
[25]	60	Pus from brain abscess	NA	NA	*Streptococcus* sp., *Prevotella* sp.	SMg	*S. constellatus*, *Prevotella* sp.
[26]	40	Pus from brain abscess	No	*S. anginosus group*, *Peptostreptococcus.*	*Mycobacterium llatzerense*, *M. immunogenum*	SMg	*Mycobacterium chubuense*, *M. abscessus* subsp. *bolletii*

Abbreviations: CSF, cerebrospinal fluid; NA, not available; and SMg, shotgun metagenomics.

## Data Availability

The data that support the findings of this case report are available from the corresponding author upon reasonable request. The sequence data are available under the accession number (PRJDB18608) in BioProject.

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
