# Peer review of "Paraclostridium tenue Causing an Anaerobic Brain Abscess Identified by Whole-Metagenome Sequencing: A Case Report"

_microorganisms, 2024, doi:10.3390/microorganisms12081692_

Round 1
Reviewer 1 Report
Comments and Suggestions for Authors
The authors reported a case report of Eubacterium tenue causing an anaerobic brain abscess identified by whole-metagenome sequencing
After careful review, I have some suggestions for improving the manuscript:
- In the abstract and introduction section, the authors need to clarify the AIM of the paper (rare condition reported? or first time identified with whole-metagenome sequencing? )
- how do the authors explain the mild hearing loss in the left ear?
- Has surgery been considered? yes or no? Why yes or why not?
- to improve the quality of the paper, a table concomitant systematic literature review (and table of similar paper) would be added
- English need a syntax and grammar revision
Comments on the Quality of English Language
- English need a syntax and grammar revisione
Author Response
Dear Reviewer 1:
We wish to thank the reviewer for their careful review of the manuscript and helpful feedback, which have improved the quality of our case report. We appreciate the opportunity to resubmit our revised manuscript.
Our responses to the reviewer’s suggestions and concerns are provided below.
Reviewer 1
- In the abstract and introduction section, the authors need to clarify the AIM of the paper (rare condition reported? or first time identified with whole-metagenome sequencing? )
>RESPONSE: Thank you for bringing this to our attention. Our paper has two primary aims: First, to introduce Paraclostridium tenue as a previously unreported pathogen of brain abscesses. Second, to advocate for the increased use of whole-metagenome sequencing in routine clinical settings. This approach can accurately identify pathogens that conventional bacterial culture tests might not be able to and contribute to achieving long-term remission. These points have been stated in the Abstract, Introduction, and Discussion sections as follows:
(Page 1, Line 25–28)
Therefore, P. tenue can be a causative pathogen for brain abscesses. Furthermore, whole-metagenome sequencing is a promising method for detecting rare pathogens that are not identifiable by conventional bacterial culture tests. This approach enables a more targeted treatment and contributes to achieving long-term remission in clinical settings.
(Page 2, Line 52–54)
Therefore, whole-metagenome sequencing is valuable for identifying pathogens and achieving long-term remission and should be more commonly employed in daily clinical settings.
(Page 4, Line 183–186)
Whole-metagenome sequencing is a promising method for detecting rare pathogens that cannot be identified by conventional bacterial culture tests commonly used in clinical settings and research, and for achieving long-term remission of brain abscesses.
- how do the authors explain the mild hearing loss in the left ear?
>RESPONSE: Thank you for highlighting this. Otitis media was not observed in the otological evaluation or brain MRI. P. tenue was detected in the cerebrospinal fluid (CSF), suggesting that it also caused bacterial meningitis. Meningitis due to P. tenue could lead to sensorineural hearing loss, which is a common sequela of bacterial meningitis, affecting up to 54% of survivors (Jensen et al., Open Forum Infect Dis. 2023). Moreover, experimental studies have demonstrated that suppurative labyrinthitis with cochlear damage can occur due to infection spreading from the subarachnoid space (Kay et al., Neuropathol Appl Neurobiol. 1991; Brandt et al., Neurobiol Dis. 2006; Cayé-Thomasen et al., Laryngoscope. 2009; Møller et al., Otol Neurotol. 2014; Møller et al., Eur Arch Otorhinolaryngol. 2015). We have included this explanation in the Discussion section as follows:
(Page 4, Line 175–181)
The patient experienced mild hearing loss. CSF analysis revealed the presence of P. tenue. This finding indicates that P. tenue in the CSF was the etiological agent responsible for bacterial meningitis in this case. Meningitis caused by P. tenue could lead to sensorineural hearing loss, a common complication of bacterial meningitis, affecting up to 54% of survivors [30]. Several experimental studies have shown that lesions can include suppurative labyrinthitis with cochlear damage and infection spread from the subarachnoid space [31–35].
- Has surgery been considered? yes or no? Why yes or why not?
>RESPONSE: Thank you for the question. A definitive diagnosis of a brain abscess is based on findings from stereotactic-guided aspiration or drainage of the brain lesion. However, we determined that biopsy and surgery were not feasible or necessary in this case. The patient experienced only mild symptoms such as fever, headache, and mild hearing loss, without consciousness disturbances or motor palsy. Additionally, we successfully identified P. tenue as the causal pathogen through shotgun sequencing of the CSF. We have added this explanation as follows:
(Page 3, Lines 117–120)
Therefore, we determined that biopsy and surgery were not feasible or necessary for the patient, as her symptoms were limited to fever, headache, and mild hearing loss, without consciousness disturbances or motor palsy. Furthermore, P. tenue was identified as the causal pathogen through shotgun sequencing.
(Page 3–4, Lines 143–149)
Generally, patients with brain abscesses may require stereotactic-guided needle aspiration or surgical drainage in addition to antibiotics for both diagnostic and therapeutic purposes. However, for this patient, these procedures were deemed too invasive due to the mild nature of her symptoms, which included only fever, headache, and mild hearing loss, without consciousness disturbances or motor palsy. Moreover, the identification of P. tenue as the causal pathogen by shotgun sequencing using CSF made these invasive procedures unnecessary, leading to a successful de-escalation from empirical therapy.
- to improve the quality of the paper, a table concomitant systematic literature review (and table of similar paper) would be added
>RESPONSE: Thank you for your suggestion. We have summarized previous reports on pathogens identified in brain abscesses through whole-metagenomic analyses. We have added a new table (Table 3) along with the explanation in the Discussion section as follows:
(Page 4, Line 154–156)
To date, whole-metagenomic analysis has identified causal pathogens in 17 cases of brain abscesses, whereas bacterial culture tests identified pathogens in only 6 of these cases (Table 3, [13–26]).
- English need a syntax and grammar revision
>RESPONSE: This manuscript has been proofread by an English language editing service. We have added this in the Acknowledgments section.
(Page 11, Line 228–229)
We thank the patient for providing informed consent and Enago (https://www.enago.com/) for the English language editing services.
Reviewer 2 Report
Comments and Suggestions for Authors
The article describes a case of brain abscess with hemorrhagic transformation. Lumbar puncture was performed and whole-metagenome sequencing was performed to identify Eubacterium tenue as a causative organism. Successful antibiotic treatment then led to complete cure.
The introduction is concise. The case report itself is well presented and will clear idea of how the proper treatment should be performed. Especially Figure 1 and 3 are very informative about the appearance of the abscess and the later follow up. The description of the actual metagenome sequence and its importance for the treatment is also well described.
I have only some concerns about the discussion where I miss a slightly broader view about other possible diagnostic procedures (for example abscess puncture and culturing). I would also recommend a short description of other treatment modalities (surgery) for brain abscess and why surgery was not indicated in this case. After those minor corrections the article is suitable to be published in the journal.
Comments on the Quality of English LanguageThe article describes a case of brain abscess with hemorrhagic transformation. Lumbar puncture was performed and whole-metagenome sequencing was performed to identify Eubacterium tenue as a causative organism. Successful antibiotic treatment then led to complete cure.
The introduction is concise. The case report itself is well presented and will clear idea of how the proper treatment should be performed. Especially Figure 1 and 3 are very informative about the appearance of the abscess and the later follow up. The description of the actual metagenome sequence and its importance for the treatment is also well described.
I have only some concerns about the discussion where I miss a slightly broader view about other possible diagnostic procedures (for example abscess puncture and culturing). I would also recommend a short description of other treatment modalities (surgery) for brain abscess and why surgery was not indicated in this case. After those minor corrections the article is suitable to be published in the journal.
Author Response
Dear Reviewer 2:
We wish to thank the reviewer for their careful review of the manuscript and helpful feedback, which have improved the quality of our case report. We appreciate the opportunity to resubmit our revised manuscript.
Our responses to the reviewer's suggestions and concerns are provided below.
Reviewer2
The article describes a case of brain abscess with hemorrhagic transformation. Lumbar puncture was performed and whole-metagenome sequencing was performed to identify Eubacterium tenue as a causative organism. Successful antibiotic treatment then led to complete cure.
The introduction is concise. The case report itself is well presented and will clear idea of how the proper treatment should be performed. Especially Figure 1 and 3 are very informative about the appearance of the abscess and the later follow up. The description of the actual metagenome sequence and its importance for the treatment is also well described.
I have only some concerns about the discussion where I miss a slightly broader view about other possible diagnostic procedures (for example abscess puncture and culturing). I would also recommend a short description of other treatment modalities (surgery) for brain abscess and why surgery was not indicated in this case. After those minor corrections the article is suitable to be published in the journal.
>RESPONSE: We agree that this is an important point. Although some patients with brain abscesses may require stereotactic-guided needle aspiration or surgical drainage in addition to antibiotics for both diagnostic and therapeutic purposes, we determined that these procedures were not feasible or necessary in our case. The procedures were considered too invasive because the patient only experienced mild symptoms, including fever, headache, and mild hearing loss, without consciousness disturbances or motor palsy. Furthermore, successful identification of P. tenue as the causal pathogen through shotgun sequencing of the CSF allowed us to de-escalate from empirical therapy. Therefore, we have added this explanation in the Results and Discussion sections as follows:
(Page 3, Lines 117–120)
Therefore, we determined that a biopsy and surgery were not feasible or necessary for the patient, as her symptoms were limited to fever, headache, and mild hearing loss, without consciousness disturbances or motor palsy. Furthermore, P. tenue was identified as the causal pathogen through shotgun sequencing.
(Page 3–4, Lines 143–149)
Generally, patients with brain abscesses may require stereotactic-guided needle aspiration or surgical drainage in addition to antibiotics for both diagnostic and therapeutic purposes. However, for this patient, these procedures were deemed too invasive due to the mild nature of her symptoms, which included only fever, headache, and mild hearing loss, without consciousness disturbances or motor palsy. Moreover, the identification of P. tenue as the causal pathogen by shotgun sequencing using CSF made these invasive procedures unnecessary, leading to a successful de-escalation from empirical therapy.
Reviewer 3 Report
Comments and Suggestions for Authors
The manuscript by Chiba et al. titled "Eubacterium tenue causing an anaerobic brain abscess identified by whole-metagenome sequencing: A case report" describes a clinical case of a brain abscess caused by a rare anaerobic bacterium. The researchers applied ribosomal MLST and whole-genome metagenomic sequencing methods. The article is interesting; however, it has several significant shortcomings.
First, the whole-genome sequencing data is not publicly available, this information about accession number is not provided in the manuscript. This is a strict requirement for authors. Second, due to the abundance of methods used, I strongly recommend creating a separate Materials and Methods section.
Additional comments are provided below:
Line 2: As far as I know, according to modern nomenclature, the correct name of the bacterium is Paraclostridium tenue. I suggest that this name be used throughout the manuscript.
Line 4: There is no need to include titles, such as MD, PhD. An asterisk should be added for the second author.
Line 6: Email addresses for all authors.
Line 22 and onwards: The Latin name of the bacterium should be italicized.
Line 28: There is no mention of shotgun sequencing in the Methods section. Should this keyword be removed?
Line 84: Could you provide more details about the isolation of the pure culture? It is unclear whether a pure culture of the microbe was isolated or not. Which colonies were sequenced? How were they selected?
Line 85: Which program and method were used for the phylogenetic analysis?
Line 89: What version of Unicycler was used?
Line 90: How exactly was the ribosomal MLST performed? What method was used? Where are the references?
Line 95: Which program was used to perform the ANI analysis?
Figure 2: Please add accession numbers in GenBank for the genomes and bootstrap values on the tree.
Line 109: Latin names should be italicized.
Table 1: What do footnotes 5) and 6) mean?
Table 2: The title is the same as that of Table 1. Similarly, please add accession numbers.
Line 124 and onward: Family/genus names should be italicized.
Line 167: Add accession number and the BioProject number.
Comments on the Quality of English LanguageQuality of English is enough for understanding.
Author Response
Dear Reviewer 3:
We wish to thank the reviewer for their careful review of the manuscript and helpful feedback, which have improved the quality of our case report. We appreciate the opportunity to resubmit our revised manuscript.
Our responses to the reviewer's suggestions and concerns are provided below.
Reviewer3
The manuscript by Chiba et al. titled "Eubacterium tenue causing an anaerobic brain abscess identified by whole-metagenome sequencing: A case report" describes a clinical case of a brain abscess caused by a rare anaerobic bacterium. The researchers applied ribosomal MLST and whole-genome metagenomic sequencing methods. The article is interesting; however, it has several significant shortcomings.
First, the whole-genome sequencing data is not publicly available, this information about accession number is not provided in the manuscript. This is a strict requirement for authors. Second, due to the abundance of methods used, I strongly recommend creating a separate Materials and Methods section.
>RESPONSE: Thank you for your suggestions.
Genome sequence data: We have deposited the genome sequence in the public database (NCBI/EMBL/DDBJ) and have included the BioProject accession number in the revised manuscript. Details of the accession number will remain private until the paper is accepted and will be made public once the paper is accepted.
(Page 11, Lines 226–227)
The sequence data are available under the accession number (PRJDB18608) in BioProject.
Material and Methods: We have added the following Materials and Methods section:
(Page 2, Lines 56–71)
2. Materials and Methods
We used a test-tube growth medium for anaerobic bacteria, and a test-tube growth medium supplemented with hemin and vitamin K₁. The enriched bacteria were then transferred to an agar medium suitable for anaerobic culture, also supplemented with hemin and vitamin K₁, and incubated at 35°C to obtain pure cultures.
To identify the causative microorganism, we performed 16S rRNA gene and whole-metagenome sequencing of the pure cultured bacteria from the cultured strains of CSF. First, we used MEGA (https://www.megasoftware.net/) for the phylogenetic analysis of the full-length 16S rRNA gene sequence. Subsequently, whole-metagenome analysis was performed using DNA extracted from cultured bacterial isolates. The genome sequence was assembled into 28 contigs by shotgun sequencing using MiSeq®️ and de novo assembly by Unicycler (ver. 0.4.4). Ribosomal multilocus sequence typing (rMLST, https://pubmlst.org/species-id) was used as well [5]. Finally, the genomic information was compared to other two strains with previously reported whole metagenomes. Additionally, we used whole-metagenome information for average nucleotide identity (ANI) analysis (https://github.com/ParBLiSS/FastANI) [6].
Additional comments are provided below:
Line 2: As far as I know, according to modern nomenclature, the correct name of the bacterium is Paraclostridium tenue. I suggest that this name be used throughout the manuscript.
>RESPONSE: Thank you for your valuable comment. As you suggested, we have made this correction throughout the manuscript. Moreover, we reanalyzed the results and revised Tables 1 and 2, and Figure 2.
Line 4: There is no need to include titles, such as MD, PhD. An asterisk should be added for the second author.
>RESPONSE: Thank you for pointing this out. We have removed titles such as MD and PhD, and an asterisk is added for the second author.
Line 6: Email addresses for all authors.
>RESPONSE: Thank you for your suggestion. We have added email addresses of all authors.
Line 22 and onwards: The Latin name of the bacterium should be italicized.
>RESPONSE: Thank you for your suggestion. The Latin name of the bacterium has been italicized.
Line 28: There is no mention of shotgun sequencing in the Methods section. Should this keyword be removed?
>RESPONSE: Thank you for pointing this out. As you suggested, the term “shotgun sequencing” has been removed from the Keywords.
Line 84: Could you provide more details about the isolation of the pure culture? It is unclear whether a pure culture of the microbe was isolated or not. Which colonies were sequenced? How were they selected?
>RESPONSE: Thank you very much for your comment. We have added the following description to the Materials and Methods section as follows:
(Page 2, Lines 57–60)
We used a test-tube growth medium for anaerobic bacteria, a test-tube growth medium supplemented with hemin and vitamin K₁. The enriched bacteria were then transferred to an agar medium suitable for anaerobic culture, also supplemented with hemin and vitamin K₁, and incubated at 35°C to obtain pure cultures.
Line 85: Which program and method were used for the phylogenetic analysis?
>RESPONSE: Thank you for your question. We used the MEGA program (https://www.megasoftware.net/) for the phylogenetic analysis. This information has been added to the Materials and Methods section.
Line 89: What version of Unicycler was used?
>RESPONSE: Thank you for your question. We used Unicycler version 0.4.4, and this information has been added to the Material and Methods section.
Line 90: How exactly was the ribosomal MLST performed? What method was used? Where are the references?
>RESPONSE: Thank you for your question. We used Pub MLST system (https://pubmlst.org/species-id), as described by Jolley et al. Microbiology. 2012 cited as reference number 5 in our revised manuscript. We have included this explanation and reference to the Materials and Methods section.
Line 95: Which program was used to perform the ANI analysis?
>RESPONSE: Thank you for your question. We used the Fast-ANI system (https://github.com/ParBLiSS/FastANI) for the ANI analysis. We have added this information to the Materials and Methods section.
Figure 2: Please add accession numbers in GenBank for the genomes and bootstrap values on the tree.
>RESPONSE: Thank you for your comment. As you suggested, we have added the GenBank accession numbers and bootstrap values on the tree in the revised Figure 2.
Line 109: Latin names should be italicized.
>RESPONSE: Thank you for your comment. We have italicized the Latin names throughout the manuscript.
Table 1: What do footnotes 5) and 6) mean?
>RESPONSE: Thank you for your comment. Footnotes 5) and 6) in the original Table 1, which indicated reference numbers, have been updated to [7] in the revised Table 1.
Table 2: The title is the same as that of Table 1. Similarly, please add accession numbers.
>RESPONSE: Thank you for bringing this to our attention. The title of Table 2 has been corrected to “Comparison of average nucleotide identity (ANI).” We have also added the accession numbers to this table.
Line 124 and onward: Family/genus names should be italicized.
>RESPONSE: Thank you for your comment. We have italicized the Latin name of the bacterium.
Line 167: Add accession number and the BioProject number.
>RESPONSE: Thank you for pointing this out. We have added the BioProject accession number (PRJDB18608) in the manuscript. The details of this accession number will be kept private until the paper is accepted; they will be made public as soon as the paper is accepted.
(Page 11, Lines 226–227)
The sequence data are available under the accession number (PRJDB18608) in BioProject.
Round 2
Reviewer 1 Report
Comments and Suggestions for Authors
Good improvmentes
Reviewer 3 Report
Comments and Suggestions for Authors
The authors have addressed the issues and considered all my comments. There are just a couple more things to add:
Line 61: I suggest to add version of MEGA and algorithm (probably, Maximum Likelihood?) and number of bootstraps.
Line 68: Add version od FastANI.
Comments on the Quality of English LanguageEnglish quality is good enough.